# Geometry-Editable and Appearance-Preserving Object Composition

## Abstract

General object composition (GOC) aims to seamlessly integrate a target object into a background scene with desired geometric properties, while simultaneously preserving its fine-grained appearance details. Recent approaches derive semantic embeddings and integrate them into advanced diffusion models to enable geometry-editable generation. However, these highly compact embeddings encode only high-level semantic cues and inevitably discard fine-grained appearance details. We introduce a Disentangled Geometry-editable and Appearance-preserving Diffusion (DGAD) model that first leverages semantic embeddings to implicitly capture the desired geometric transformations and then employs a cross-attention retrieval mechanism to align fine-grained appearance features with the geometry-edited representation, facilitating both precise geometry editing and faithful appearance preservation in object composition. Specifically, DGAD builds on CLIP/DINO-derived and reference networks to extract semantic embeddings and appearance-preserving representations, which are then seamlessly integrated into the encoding and decoding pipelines in a disentangled manner. We first integrate the semantic embeddings into pre-trained diffusion models that exhibit strong spatial reasoning capabilities to implicitly capture object geometry, thereby facilitating flexible object manipulation and ensuring effective editability. Then, we design a dense cross-attention mechanism that leverages the implicitly learned object geometry to retrieve and spatially align appearance features with their corresponding regions, ensuring faithful appearance consistency. Extensive experiments on public benchmarks demonstrate the effectiveness of the proposed DGAD framework.

## 1 Introduction

General Object composition (GOC) involves editing a target object to seamlessly integrate it into a background scene at arbitrary angles and positions, while preserving the object's appearance details without alteration. A robust generative object composition system should automate complex tasks such as interactive image editing, virtual environment creation, and content generation for augmented and virtual reality (AR/VR) applications, by simultaneously supporting flexible object editing and faithful appearance preservation — a critical balance that minimizes the need for manual adjustment in both geometric editing and visual consistency. Thanks to the emergence of large-scale pre-trained diffusion models Nichol & Dhariwal (2021); Song et al. (2020); Ho & Salimans (2022), substantial progress has been made in generative compositing. However, these models still face challenges in simultaneously editing the target object according to the user's expectations and maintaining consistent appearance details, limiting their applicability in real-world scenarios.

Existing methods either introduce CLIP/DINO-derived semantic embeddings Song et al. (2023); Chen et al. (2024); Yang et al. (2023), or leverage pixel-aligned appearance features to achieve object composition Zhao (2024), as shown in Fig. 1. However, both approaches fail to simultaneously ensure object editability and appearance-preserving. The former encodes the object into a compact semantic embedding, which demonstrates strong compatibility with pre-trained diffusion models and enables robust manipulation of geometric properties such as shape deformation and viewpoint changes. Yet, this compact encoding inevitably loses high-frequency details, making it difficult to preserve appearance during editing. In contrast, the latter employs reference networks to extract pixel-wise appearance features, which maintain tight spatial correspondence with the object and ex-

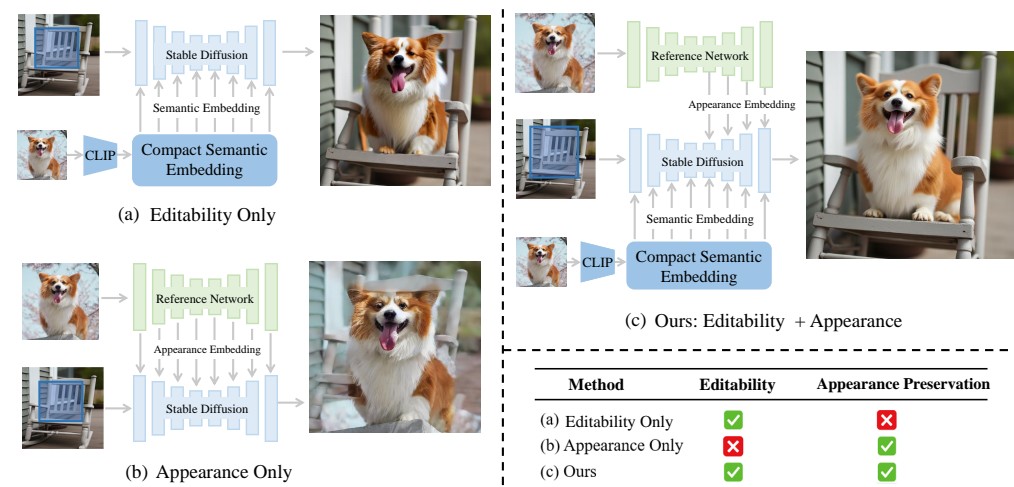

Figure 1: (a) Leverages compact semantic embeddings to enable object editability but fails to preserve appearance details. (b) Utilizes appearance features to retain visual fidelity, yet lacks editing capability. Unlike both, Our method implicitly learns the geometry-editable representation and explicitly aligns fine-grained appearance features with the geometry-edited representation, facilitating both precise geometry editing and faithful appearance preservation.

cel at reconstructing appearance details from noise latents. Nevertheless, the rigid spatial alignment severely restricts editing flexibility, often resulting in copy-paste-like outputs rather than adaptive transformations. Object composition requires both object editability and appearance-preserving, and designing an effective mechanism to leverage the advantages of both is key to solving the problem.

To address these challenges, we first leverage semantic embeddings to implicitly capture the desired geometric transformations, and then employ a dense cross-attention retrieval mechanism to align fine-grained appearance features with the geometry-edited representation. Prior methods—such as Ye et al. (2023); Yang et al. (2023), which rely on precise object masks to explicitly encode geometric properties, thereby limiting editing flexibility and efficiency, or Zhao (2024), which performs implicit retrieval and alignment of appearance features via a cross-attention mechanism—often result in suboptimal appearance preservation due to inconsistent geometry-appearance mapping. In contrast, we leverage the inherent spatial reasoning capabilities of pretrained diffusion models to implicitly capture object geometry, and subsequently perform position-wise retrieval and alignment of appearance features with their corresponding geometric regions based on the captured geometry, ensuring both editability and appearance preservation during composition.

To this end, we propose the Disentangled Geometry-editable and Appearance-preserving Diffusion (DGAD) model, which leverages compact semantic embeddings to implicitly learn the geometric properties of objects during encoding, and employs the resulting representations to explicitly retrieve and spatially map appearance features to corresponding geometric regions during decoding. Specifically, during the encoding stage, the initial input is constructed by concatenating the user-specified regions with surrounding contextual information. We leverage CLIP/DINO-derived semantic embeddings Radford et al. (2021); Oquab et al. (2023) and a cross-attention mechanism built upon the strong spatial reasoning capabilities of pretrained diffusion models to implicitly learn the geometric properties of objects, which in turn enables flexible manipulation and thereby ensures object editability. To ensure consistent object appearance, we introduce a dense cross-attention mechanism that leverages encoded features to establish explicit correspondences with appearance features from a reference network. The encoded features, which capture both semantic and geometric properties, serve as queries, while the appearance features act as keys and values. To further guide the attention toward semantically relevant regions, a position-wise gating weight is learned from the query features to explicitly represent the object's geometric structure, enabling the model to adaptively retrieve and align appearance features with their corresponding geometric regions. This mechanism is applied exclusively during the decoding stage, where appearance retrieval is conditioned on fully geometry-edited representation, thereby enhancing both object editability and appearance-preserving in the composition process.

Our contributions are threefold. First, we propose the Disentangled Geometry-editable and Appearance-preserving Diffusion (DGAD) model, a novel framework that is the first to explicitly disentangle geometry editing from appearance preservation in object composition. It achieves this by implicitly learning geometry during encoding and explicitly retrieving appearance features during decoding. Second, we introduce a dense attention mechanism that establishes position-aware correspondences between the edited geometry and original appearance, ensuring high-fidelity preservation under complex transformations. Finally, extensive experiments demonstrate that DGAD significantly outperforms state-of-the-art methods in both geometry editability and appearance preservation. Our code and models will be made publicly available.

## 2 RELATED WORK

The longstanding challenge in image generation is to achieve both geometric editability and visual consistency Chen et al. (2025); Xu et al. (2025), a problem particularly acute in object composition. This section reviews prior work, highlighting a persistent trade-off between these two competing goals.

Early works approached object composition primarily as an image harmonization Guerreiro et al. (2023); Jiang et al. (2021); Ke et al. (2022); Xue et al. (2022) or blending Pérez et al. (2023); Wu et al. (2019); Zhang et al. (2021) task. These methods, such as the state-of-the-art DCCF Xue et al. (2022), excel at matching color and lighting but entirely neglect geometric alignment. While subsequent GAN-based methods like GauGAN Park et al. (2019) began to address geometric inconsistencies, they were often limited by their reliance on segmentation maps or struggled with complex, out-of-plane transformations Zhan et al. (2019); Azadi et al. (2020).

The advent of diffusion models Nichol & Dhariwal (2021); Song et al. (2020); Ho & Salimans (2022) has significantly advanced the field, yet the core trade-off remains. Initial techniques like DreamBooth Ruiz et al. (2023) and Textual Inversion Gal et al. (2022) offered high-fidelity object customization but lacked direct control over geometry. To address this, a major research direction has emerged that uses compact semantic embeddings for editability. Methods like Pbe Yang et al. (2023), ObjectStitch Song et al. (2023), and IMPRINT Song et al. (2024) leverage CLIP/DINO embeddings to enable flexible geometric manipulation. However, the highly compressed nature of these embeddings inevitably discards fine-grained details, creating a core conflict between editability and appearance preservation.

Subsequent works have attempted to bridge this gap. AnyDoor Chen et al. (2024) improves fidelity by incorporating structured representations (e.g., Canny edges), but this reliance on rigid inputs constrains its flexibility for non-rigid objects. More recently, MiniBrush Zhao (2024) achieves strong appearance retention using a reference network but lacks a robust mechanism to explicitly align appearance features with the edited geometry, leading to mismatches in complex compositions. In contrast, our DGAD framework leverages semantic embeddings for implicit geometry editing while introducing a novel dense attention mechanism to explicitly and robustly align appearance features, thereby achieving both goals simultaneously.

## 3 METHOD

The overall pipeline of the DGAD framework is illustrated in Fig. 2. In the encoding stage, we utilize CLIP/DINO-derived semantic embeddings and a cross-attention mechanism built upon the strong spatial reasoning capabilities of pretrained diffusion models to implicitly capture object geometry. During decoding, a dense cross-attention mechanism is employed to explicitly retrieve and positionally align appearance features with their corresponding geometric regions based on the encoded features. In the following sections, we first describe the learning process of the geometry-editable representation (Section 3.1), then detail the appearance-preserving representation retrieval and learning (Section 3.2), and finally present the optimization procedure of the proposed DGAD framework (Section 3.3).

### 3.1 GEOMETRY-EDITABLE ENCODER

This section details the learning process for capturing implicit geometric properties of objects to enhance editing capabilities. While current approaches combining semantic embeddings Radford et al. (2021); Oquab et al. (2023) with pretrained diffusion models enable basic geometric editability, existing integration strategies remain limited in both flexibility and effectiveness. Some methods Ye et al. (2023); Yang et al. (2023) formulate the GOC task as image inpainting and attempt to model

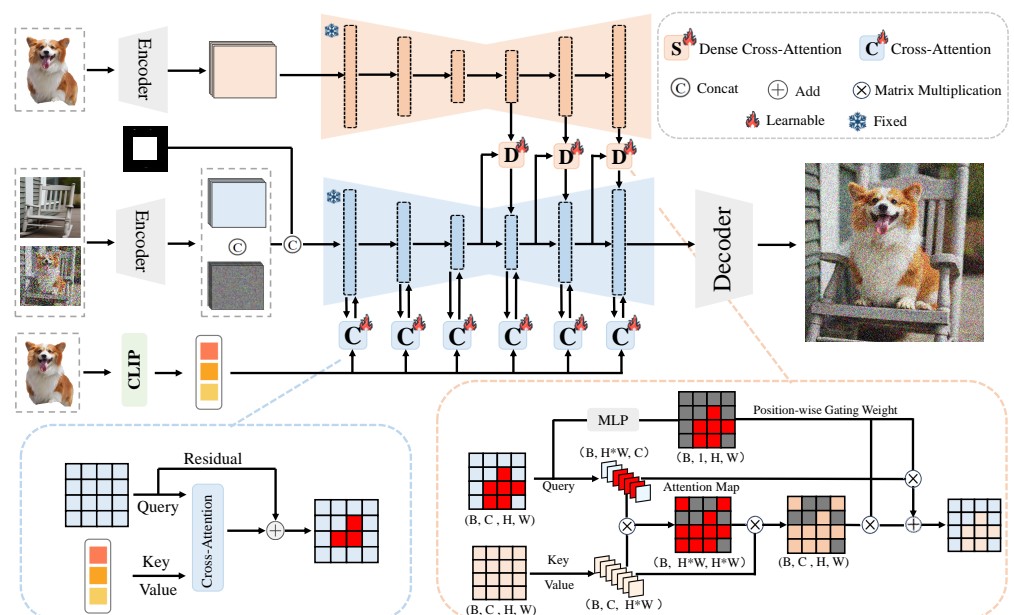

Figure 2: The training process of the proposed Disentangled Geometry-editable and Appearance-preserving Diffusion (DGAD) is as follows (the inference process is similar but involves iterative denoising): It first leverages semantic embeddings to implicitly capture the desired geometric transformations, and then employs a cross-attention retrieval mechanism to align fine-grained appearance features with the geometry-edited representation, facilitating both precise geometry editing and faithful appearance preservation in object composition.

object geometry through end-to-end training, but their practical application is constrained by dependence on fine-grained masks. Alternative approaches Chen et al. (2024) employ ControlNet-based architectures conditioned on specified regions but fall short in fully capturing geometric structure due to architectural limitations. Inspired by recent work Su et al. (2025) that demonstrates introducing an intermediate 2D layout representation greatly strengthens the geometric perception and reasoning of the generative model, we propose directly concatenating user-specified regions with surrounding contextual information as initial input, thereby utilizing the pretrained diffusion model's geometric perception ability to simplify the learning process.

Given paired training data $(I_{\text{obj}}, I_{\text{bg}}, M, I_{\text{tgt}})$, our process begins by encoding the object $I_{\text{obj}}$ into a compact semantic feature $f_{\text{obj}}$ using a pre-trained CLIP/DINO encoder. Concurrently, the background $I_{\text{bg}}$ and target $I_{\text{tgt}}$ images are encoded into latent representations, $f_{\text{bg}}$ and $f_{\text{tgt}}$, via the VAE of the pretrained diffusion model. The mask $M$ is downsampled to match the latent space resolution. To simulate diffusion, we corrupt $f_{\text{tgt}}$ with noise $\epsilon$ at timestep $t$ to obtain the noisy latent $f_{\text{tgt}}^t$.

Next, we form a multi-channel input by concatenating the downsampled mask $M$, the background latent $f_{\text{bg}}$, and the noisy target $f_{\text{tgt}}^t$. Here, $M$ serves as a spatial prior to activate the diffusion model's inherent geometric perception. To fully leverage this prior, we replicate the model's original input layer weights for the expanded channels, avoiding random initialization. Subsequently, a cross-attention mechanism implicitly learns the object's geometry-editable representation. The backbone feature $f_b$ acts as the query, while the semantic embedding $f_{\text{obj}}$ serves as the key and value:

$$Q = f_b W_q, \quad K = f_{\text{obj}} W_k, \quad V = f_{\text{obj}} W_v \tag{1}$$

$$\text{Attention}(Q, K, V) = \text{softmax}\left(\frac{QK^T}{\sqrt{d_k}}\right) V \tag{2}$$

where $W_q, W_k, W_v$ are learnable projection matrices and $\sqrt{d_k}$ is a scaling factor. The model is trained by minimizing the difference between the true and predicted noise, $\epsilon$ and $\epsilon_\theta$, based on the standard diffusion objective:

$$L = \mathbb{E}_{f_{\text{tgt}}, \epsilon \sim \mathcal{N}(0, I), c, t} \left[ \left\| \epsilon - \epsilon_\theta \left( f_{\text{tgt}}^t, c, t \right) \right\|^2 \right] \tag{3}$$

where the conditioning $c$ includes $f_{\text{obj}}$, $M$, and $f_{\text{bg}}$. This process enables the trained model to capture object geometry from simple region specifications, thereby facilitating flexible editing.

## 3.2 APPEARANCE-PRESERVING DECODER

The learned geometry-editable representation fails to preserve object appearance fidelity, primarily due to semantic embeddings predominantly encode high-level semantic cues while neglecting fine-grained visual details. Recent advances Ju et al. (2024); Hu (2024) have demonstrated that appearance features extracted from reference networks can effectively reconstruct objects from noisy latents—for instance, BrushNet Ju et al. (2024) trained on open-domain data achieves direct object reconstruction using only a single reference image. Building on these insights, we introduce a dense cross-attention mechanism that explicitly retrieves and positionally aligns appearance features with their corresponding geometric regions based on geometry-edited representation, thereby ensuring both precise geometry editing and faithful appearance preservation.

Specifically, we align the geometry-edited backbone feature $f_b \in \mathbb{R}^{B \times C \times H \times W}$ from the encoder with the fine-grained appearance feature $f_r \in \mathbb{R}^{B \times C \times H \times W}$ from the reference network (BrushNet). We designate the reshaped $f_b$ as the query ($Q$) and $f_r$ as the key ($K$) and value ($V$):

$$Q = f_b W_q, \quad K = f_r W_k, \quad V = f_r W_v \tag{4}$$

Instead of relying on standard cross-attention for implicit retrieval, we introduce a **dense attention mechanism** that explicitly gates the flow of appearance information based on the learned geometry. At its core, this mechanism learns a position-wise softmask, $\alpha$, directly from the query $Q$, which spatially identifies the object's regions. Its complement, $\beta$, marks the background.

$$\alpha = \sigma(F(Q)) \in [0,1]^{B \times 1 \times H \times W}, \quad \beta = \text{mask\_process}\,(1 - \alpha) \tag{5}$$

where $F$ is a lightweight MLP with a sigmoid activation $\sigma$. These masks enable a spatially controlled fusion of appearance and geometry:

$$\text{Dense\_Attention}(Q, K, V, \alpha, \beta) = \left( \text{softmax}\left( \frac{QK^T}{\sqrt{d_k}} \right) V \right) \odot \alpha + Q \odot \beta \tag{6}$$

This formulation has a dual effect: (1) The $\odot \alpha$ term ensures that retrieved appearance features are applied *only* within the object's geometric boundaries. (2) The $\odot \beta$ term acts as a residual connection, preserving the original backbone features $Q$ (containing scene context from the encoder) in the background. The mask\_process($\cdot$) function further refines $\beta$ during training, encouraging the preservation of encoder semantics while simultaneously forcing the retrieval of appearance details from the reference network. This explicit gating mechanism establishes a dense correspondence between $f_b$ and $f_r$, ensuring high-fidelity appearance preservation.

## 3.3 OPTIMIZATION

DGAD is trained end-to-end based on the pretrained Stable Diffusion v1.5 model. A cross-attention mechanism is applied to every block in the backbone to learn the geometry-editable representation. In contrast, the dense cross-attention mechanism is applied only in the decoder stage to explicitly retrieve and spatially align appearance-preserving features with their corresponding geometric regions, conditioned on the fully learned editable representation. The final loss is defined as follows:

$$L = \mathbb{E}_{f_{tgt}, \epsilon \sim \mathcal{N}(0,I), t} \left[ \left\| \epsilon - \epsilon_\theta \left( f_{tgt}^t, [M, f_{bg}, f_{obj}, f_r], t \right) \right\|^2 \right] \tag{7}$$

This optimization updates learnable parameters via diffusion-constrained backpropagation, enabling simultaneous precision in geometry manipulation and visual consistency maintenance during object composition. *For more detailed network architecture and training details, please refer to the appendix A.1.*

## 4 EXPERIMENTS

### 4.1 EVALUATION BENCHMARK

**Competing Algorithms .** We compare our method with several recent approaches in the field of object composition, providing a basis for evaluating the effectiveness of our proposed framework. 1) AnyDoor Chen et al. (2024) (CVPR 2024): Utilizes semantic features extracted from DINO and structured representation obtained via ControlNet to perform object composition. 2) MimicBrush Zhao (2024) (NeurIPS 2024): Leverages appearance features from a reference network to guide object composition. 3) Pbe Yang et al. (2023)(CVPR 2023): Employs CLIP-derived semantic features and strong data augmentations to achieve object composition. 4) ObjectStitch Song et al. (2023)

Table 1: Quantitative comparison with prior works. Metrics are grouped by functionality: editability, appearance preservation, and semantic consistency. Our method achieves superior performance across all categories.

| Metrics | Editability | | Appearance Preservation | | Semantic Consistency | |
|---|---|---|---|---|---|---|
| Methods | IR ↑ | FID ↓ | LPIPS ↓ | DISTS ↓ | CLIP Score ↑ | DINO Score ↑ |
| Ipadapter | 42.56 | 31.35 | 18.47 | 23.46 | 87.78 | 65.09 |
| Objectstitch | 41.21 | 32.25 | 18.96 | 23.53 | 87.06 | 65.09 |
| Pbe | 43.25 | 32.07 | 20.49 | 26.48 | 85.30 | 62.14 |
| Mimicbrush | 44.88 | 30.69 | 15.33 | 19.20 | 89.34 | 69.98 |
| Anydoor | 44.81 | 26.08 | 15.82 | 19.21 | 88.21 | 69.22 |
| **Ours** | **61.14** | **15.04** | **14.94** | **18.53** | **89.38** | **69.92** |

(CVPR 2023): Introduces a content adaptor to preserve both categorical semantics and object appearance during composition. 5) IP-Adapter Ye et al. (2023) : Formulates object composition as an inpainting task, using CLIP-derived semantic features and end-to-end fine-tuning for generation.

**Dataset .** Following the approach in AnyDoor Chen et al. (2024), we use a training set consisting of 386k images and 23k video samples. For the image dataset, we apply LaMa Suvorov et al. (2022)to remove foreground objects, creating paired data that includes the object, the background scene, and the corresponding target image. For the video dataset, we also adopt AnyDoor's preprocessing method to construct triplets with the same structure as the image data. For evaluation, we select 30 object concepts from DreamBooth Ruiz et al. (2023) as test subjects. As background scenes, we manually choose 80 geometrically annotated images from the COCO-Val set Lin et al. (2014), resulting in 2,400 synthesized samples covering all combinations of the selected objects and scenes.

**Metrics .** The purpose of object composition is to edit objects with desired geometric properties to align with the background scene while preserving their appearance details. **To evaluate the object Editability**, we introduce three metrics: IR, and FID. IR is text-to-image evaluation models trained on large-scale datasets that reflect human preferences for generated images. Since high-quality object edits tend to produce more visually appealing results that align with human expectations, it serve as human-aligned indicators of editing success. FID assesses the distributional similarity between the composed images and real-world images, providing an objective measure of compositional realism. **To evaluate appearance consistency**, we introduce two metrics: LPIPS and DISTS. These metrics can robustly assess an object's appearance consistency even under geometric misalignment by measuring differences in deep feature space. Additionally, we introduce CLIP Score and DINO Score **to measure the semantic consistency** between objects as a supplement, similar to Chen et al. (2024).

4.2 QUANTITATIVE COMPARISON

In this section, we present a quantitative comparison between DGAD and other advanced methods as shown in Tab.1, and provide a detailed analysis of the underlying causes behind each method's performance.

**IPAdapter Ye et al. (2023), ObjectStitch Song et al. (2023),** and Pbe Yang et al. (2023) rely on CLIP semantic embeddings to capture object geometry, enabling moderate editability. However, the loss of high-frequency details in these embeddings hampers appearance preservation, resulting in poor performance on appearance preservation metrics such as LPIPS and DISTS. Their inability to seamlessly integrate realistic objects also leads to weaker scores in editability metrics like IR and FID. **AnyDoor** Chen et al. (2024) integrates structured representations (e.g., Canny edges) to enhance appearance preservation. However, these representations capture only contour-based appearance, and the rigid structure limits the model to basic geometric edits. As a result, it performs only average on object editability and appearance preservation metrics. **MimicBrush** Zhao (2024) extracts pixel-level appearance features via a reference network and performs implicit geometric alignment. However, lacking explicit correspondence modeling, it struggles with accurate feature mapping, leading to suboptimal composition quality. In contrast, **DGAD** first leverages the inherent spatial reasoning capabilities of pretrained diffusion models to capture the desired geometric transformations, achieving superior results on editability metrics. It then employs a cross-attention retrieval mechanism to align fine-grained appearance features with the geometry-edited representation, thereby also outperforming existing methods on appearance preservation metrics. Regarding

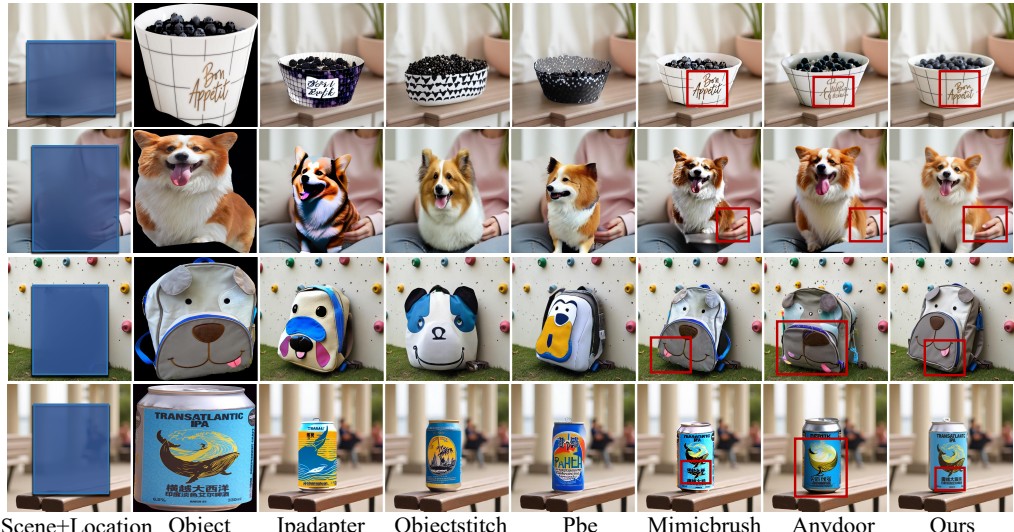

Figure 3: Qualitative comparison with recent advanced methods. The results show that the proposed method can edit objects with desired geometric properties to align with the background scene while preserving their appearance details. *For more detailed and comprehensive visualizations, including results on diverse object categories and cross-view compositions, please refer to Appendix A.2.*

semantic consistency-related metrics, existing methods all rely on CLIP/DINO-derived semantic embeddings to extract the semantic information of objects, resulting in comparable performance across these methods.

### 4.3 QUALITATIVE COMPARISONS

In this section, we present a qualitative comparison between DGAD and other advanced methods, as shown in Fig. 3, and provide a detailed analysis of the causes behind each method displayed.

**IPAdapter, ObjectStitch, and Pbe** rely on CLIP-derived semantic embeddings to learn the geometric properties of the target object for editability. However, since CLIP primarily captures category-level semantic features, it inevitably loses high-frequency visual details during the encoding process. As shown in Columns 3 to 5 of Fig. 3, while these methods can edit object geometry, they often fail to preserve appearance consistency. **AnyDoor** enhances editability and appearance preservation through DINO-based semantic embeddings combined with structured representations (e.g., Canny edges). However, its reliance on rigid structural constraints limits editing to basic geometric operations like in-plane rotations (Fig. 3, Row 2, Column 7), while incomplete appearance modeling causes visual inconsistencies. **MimicBrush** extracts pixel-level appearance features and learns object geometry from videos, achieving better appearance retention. However, due to the lack of explicit correspondence modeling, it often suffers from geometry-appearance mismatches, especially under complex edits (Fig. 3, Row 3, Column 6). In contrast, **DGAD** implicitly learns the geometry-editable representation and explicitly aligns fine-grained appearance features with the geometry-edited representation, facilitating both precise geometry editing and faithful appearance preservation, as shown in the last column of the Fig. 3.

### 4.4 USER STUDY

We conducted a user study to compare our method with other existing approaches. A total of 25 participants evaluated 30 groups of images, each containing a scene, a specified location or mask, and results generated by various methods. Participants assessed these results based on two criteria: **Composition Quality**, which measures whether the inserted object aligns with the intended geometric properties within the scene, and **Visual Consistency**, which evaluates whether the inserted object maintains consistency with the original in terms of texture, color, and material. As shown in Tab. 2, our proposed method is preferred by users on both composition quality and visual consistency metrics, demonstrating its superiority in editing objects to align geometrically with the background scene while preserving their appearance details.

Table 2: User study on the comparison between our DGAD and existing methods.

| Metric / Method | Ipadapter | Objectstitch | Pbe | Mimicbrush | Anydoor | Ours |
|---|---|---|---|---|---|---|
| Composition Quality | 12.50% | 4.17% | 0.00% | 29.17% | 16.67% | 37.50% |
| Visual Consistency | 4.17% | 4.17% | 12.50% | 21.00% | 16.67% | 41.50% |

## 4.5 ABLATION STUDY

To comprehensively evaluate the effectiveness of our proposed DGAD framework, we conduct a series of ablation studies. We begin by validating the necessity of our core disentangled design, followed by detailed analyses of the specific components within our geometry-editable encoder and appearance-preserving decoder. All results are summarized in Table 3.

Table 3: Comprehensive ablation studies of the DGAD framework. We analyze the core components, the encoder design, and the decoder design. Removing any key component or strategy leads to a notable performance degradation, demonstrating the effectiveness of our integrated design. Best results are highlighted.

| Method | IR ↑ | FID ↓ | LPIPS ↓ | DISTS ↓ | CLIP Score ↑ | DINO Score ↑ |
|---|---|---|---|---|---|---|
| *(1) Analysis of Core Framework Components* | | | | | | |
| Ours (w/o Semantic Guidance) | 35.16 | 31.55 | 15.10 | 18.98 | 86.50 | 67.23 |
| Ours (w/o Appearance Features) | 55.24 | 21.37 | 28.15 | 33.40 | 89.10 | 69.55 |
| *(2) Analysis of Geometry-Editable Encoder Design* | | | | | | |
| Ours (w/o Layout Representation) | 52.82 | 26.08 | 16.02 | 19.97 | 87.12 | 68.02 |
| Ours (w/o Copied Weights) | 58.12 | 24.67 | 15.13 | 19.32 | 88.12 | 68.18 |
| *(3) Analysis of Appearance-Preserving Decoder Design* | | | | | | |
| Ours (w/o Dense Attention) | 58.12 | 19.18 | 16.92 | 21.93 | 88.23 | 69.12 |
| Ours (Dense Attention on Both Stages) | 60.02 | 16.13 | 15.23 | 19.21 | 89.13 | 69.12 |
| **Ours (Full Model)** | **61.14** | **15.04** | **14.94** | **18.53** | **89.38** | **69.92** |

### 4.5.1 EFFECTIVENESS OF THE DISENTANGLED FRAMEWORK

**Analysis of Core Components.** We first validate the fundamental motivation behind our disentangled framework by ablating its two pillars: semantic-guided geometry control and appearance feature preservation. As shown in Table 3 (rows 1-2), we test two variants: 1) **Ours (w/o Semantic Guidance)**, which removes the CLIP/DINO embeddings and relies solely on the appearance network. 2) **Ours (w/o Appearance Features)**, which removes the reference network and uses only semantic guidance.

The results reveal the critical trade-off that DGAD is designed to solve. The model without semantic guidance maintains strong appearance fidelity (LPIPS of 15.10), closely approaching the full model's performance in this aspect. However, it fails catastrophically in geometric editing, with its IR score plummeting to 35.16, as its rigid feature mapping prevents flexible manipulation. Conversely, the model without appearance features achieves strong editability and semantic consistency, with scores second only to our full model. Yet, it completely fails to preserve the object's visual identity, leading to extremely poor appearance scores (LPIPS of 28.15). This clearly illustrates the central conflict of the GOC task. Our full model is the only configuration that excels in both domains simultaneously, achieving the best performance across all metrics and confirming the strong synergistic effect of our disentangled design.

### 4.5.2 ANALYSIS OF THE GEOMETRY-EDITABLE ENCODER

Having established the necessity of the core components, we now analyze the specific design choices within the encoder.

**Effectiveness of Geometric Prior.** In Section 3.1, we introduce a 2D layout representation to activate the geometric perception capabilities of the pretrained diffusion model. When this component is removed ("Ours w/o Layout Representation"), the model degrades to a standard inpainting setup. Table 3 shows a significant drop in editability metrics (e.g., FID worsens from 15.04 to 26.08). This demonstrates that without explicit layout guidance, the model struggles to leverage its inherent

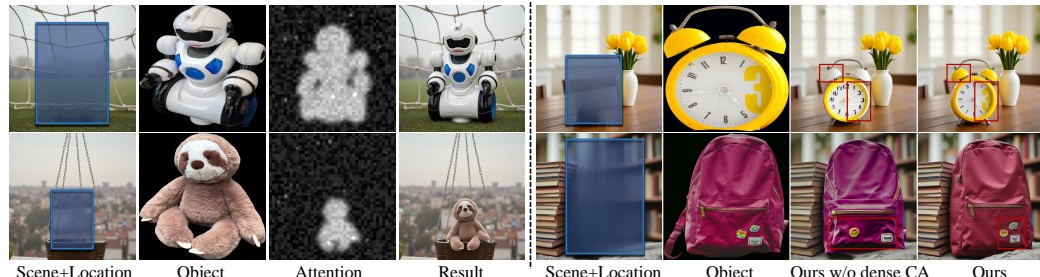

| Scene+Location | Object | Attention | Result | Scene+Location | Object | Ours w/o dense CA | Ours |

Figure 4: **Left half:** Visualization of the implicitly captured geometric properties of the object. **Right half:** Qualitative comparisons of "Ours w/o dense CA" and "Ours", Using dense cross-attention effectively retrieves accurate appearance features, thereby promoting appearance preservation.

geometric priors, leading to weaker shape understanding and degraded composition quality. The attention map visualization in Fig. 4 (left) further confirms that our layout-guided approach enables precise, user-specified geometric control.

**Effectiveness of Weight Initialization.** We argue that copying pretrained weights for the expanded input channels is crucial for knowledge retention. To verify this, we compare against a baseline with randomly initialized weights ("Ours w/o Copied Weights"). The results in Table 3 show a notable performance drop, particularly in FID (24.67 vs. 15.04). This confirms that weight inheritance effectively transfers rich geometric knowledge from the pretrained model, a task for which learning from scratch on a smaller dataset is insufficient.

### 4.5.3 ANALYSIS OF THE APPEARANCE-PRESERVING DECODER

Finally, we ablate the key designs within our appearance-preserving decoder.

**Effectiveness of Dense Cross Attention.** Our core novelty in the decoder is the Dense Cross Attention mechanism. When we replace it with standard cross-attention ("Ours w/o Dense Attention"), Table 3 shows a marked decline in appearance preservation, with LPIPS increasing from 14.94 to 16.92 and DISTS from 18.53 to 21.93. This is because the standard mechanism lacks the capacity for explicit, position-aware feature retrieval, struggling to map fine-grained details to their correct geometric regions. The visual artifacts in Fig. 4 (right) corroborate this deficiency.

**Effectiveness of Optimization Strategy.** We apply Dense Attention only at the decoder stage. To validate this strategy, we test a variant where it is applied at both encoder and decoder stages ("Ours (Dense Attention on Both Stages)"). As shown in Table 3, this approach slightly degrades appearance preservation. We attribute this to the noisy inputs during the early diffusion steps in the encoder; forcing appearance retrieval based on noisy latents can introduce erroneous visual cues and contaminate the final output. Confining Dense Attention to the decoder, where geometric representations are more stable, proves to be a more robust and effective strategy.

## 5 CONCLUSION

In this work, we introduced the Disentangled Geometry-editable and Appearance-preserving Diffusion (DGAD) model, a novel framework designed to address a central challenge in general object composition: simultaneously achieving precise geometric control without sacrificing fine-grained appearance details. Our key insight lies in disentangling these two competing objectives. Instead of relying on a single, overloaded representation, DGAD employs a two-stage, synergistic process. During the encoding phase, we leverage the powerful spatial reasoning of pre-trained diffusion models, guided by compact semantic embeddings, to implicitly learn the desired geometric transformations. This ensures high-level editability and flexible object manipulation. Subsequently, in the decoding phase, we introduce a dense cross-attention mechanism. This mechanism critically uses the geometry captured during encoding as a spatial guide to retrieve and meticulously align high-fidelity appearance features onto their corresponding regions. This disentangled architecture ensures that geometric edits do not corrupt the object's intrinsic appearance, leading to a highly controllable and faithful composition process. Extensive experiments on public benchmarks validate the superiority and effectiveness of our DGAD framework, marking a significant step towards more practical and high-quality object composition.

## ETHICS AND REPRODUCIBILITY STATEMENT

### ETHICS STATEMENT

The proposed DGAD framework is designed for creative applications, such as interactive image editing and content creation for virtual environments. We acknowledge that, like all generative models, it has the potential for misuse in creating misleading or harmful content. However, our work does not introduce any new intrinsic risks beyond the scope of existing generative technologies. We have conducted our research using publicly available datasets (e.g., COCO, DreamBooth datasets) to ensure no privacy violations. We advocate for the responsible use of this technology and encourage the development of robust detection methods for synthetic media as a community-wide effort.

### REPRODUCIBILITY STATEMENT

To ensure the reproducibility of our results, we commit to making our code, pre-trained models, and evaluation scripts publicly available upon publication. Our framework is built upon well-established open-source libraries, including PyTorch and the Hugging Face `diffusers` library. The datasets used for training and evaluation are publicly accessible benchmarks, with details and preprocessing steps described in Section 4.1. Furthermore, we provide comprehensive implementation details, including network architecture, training hyperparameters, and computational environment specifics, in the Appendix (Section A.1). This information should be sufficient for the research community to reproduce our findings and build upon our work.

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

## A  APPENDIX

Due to the page limit in the main manuscript, we provide additional implementation details and experimental results in this supplementary material to enhance reproducibility and completeness. This supplement covers the following aspects: 1) We present the network architecture and training details of DGAD for better reproducibility. 2) We provide more visualization results and extensive ablation studies for a more comprehensive comparative analysis. 3) We also submit our code as a zipped supplementary file. Since the training and testing data are publicly available datasets, we provide relevant links to facilitate reproduction.

### A.1  IMPLEMENTATION DETAILS

**Training Details .** Our model was trained on four NVIDIA RTX 4090 GPUs with a batch size of 2 per GPU, resulting in an effective batch size of 8. The training process lasted approximately four days. We utilized the Adam optimizer Kingma (2014) with a learning rate of $1 \times 10^{-5}$. To enhance memory efficiency and computational speed, mixed precision training (fp16) was employed. The training was conducted using the `accelerate` library, incorporating memory-efficient attention via the xformers backend. Input images were resized to a resolution of $512 \times 512$. During inference, all methods were evaluated using a classifier-free guidance (CFG) scale of 7.5 and 50 denoising steps.

**Network Details .** We adopt the inpainting version of Stable Diffusion v1.5 Stacchio (2023) as our backbone model. This choice is made primarily to ensure a fair and direct comparison with the majority of prior works we evaluate against, which are also built upon the SD v1.5 architecture.This allows us to transparently benchmark the specific contributions of our proposed modules. We select the corresponding pretrained BrushNet Ju et al. (2024) as the reference network. Since the Stable Diffusion inpainting backbone already possesses the ability to generate content conditioned on text prompts, we keep both the backbone and BrushNet fixed during training. We only train the additionally introduced standard cross-attention module to learn the geometric properties of the object, as well as the dense cross-attention module to retrieve and align appearance features with their corresponding geometric regions. For a clearer and more intuitive understanding, we present the pseudocode of the Dense Cross Attention in Algorithm 1.

**Algorithm 1** Dense Cross Attention

---

**Require:** Feature maps $f_b, f_r \in \mathbb{R}^{B \times C \times H \times W}$
**Ensure:** Output feature map $output \in \mathbb{R}^{B \times C \times H \times W}$
 1: **function** DENSECROSSATTENTION($f_b, f_r$)
 2: $\quad q \leftarrow \text{Conv1x1}(f_b)$
 3: $\quad k \leftarrow \text{Conv1x1}(f_r)$
 4: $\quad v \leftarrow \text{Conv1x1}(f_r)$
 5: $\quad$ **for** $l = 1$ **to** $L$ **do**
 6: $\quad\quad q \leftarrow \text{Conv3x3}(q)$
 7: $\quad\quad k \leftarrow \text{Conv3x3}(k)$
 8: $\quad\quad v \leftarrow \text{Conv3x3}(v)$
 9: $\quad$ **end for**
10: $\quad \alpha \leftarrow \text{Sigmoid}(\text{MLP}(q))$
11: $\quad \beta \leftarrow \text{MaskProcess}(1 - \alpha)$
12: $\quad$ Reshape $q, k, v$ to $[B, HW, C]$
13: $\quad A \leftarrow \text{softmax}\left(q \cdot k^\top / \sqrt{C}\right)$
14: $\quad attn \leftarrow A \cdot v$
15: $\quad$ Reshape $attn$ to $[B, C, H, W]$
16: $\quad output \leftarrow attn \odot \alpha + q \odot \beta$
17: $\quad$ **return** $output$
18: **end function**
19: **function** MASK_PROCESS($M$)
20: $\quad M_{clamped} \leftarrow \text{clamp}(M, 0.5, 0.8)$
21: $\quad$ **return** $M_{clamped}$
22: **end function**

---

## A.2 EXPERIMENTS

**Qualitative Comparisons .** Due to page limitations, this appendix provides extensive qualitative results to further validate the effectiveness of our proposed method.

First, we showcase the generalization of our approach across a diverse set of object categories and scenes in Figs. 5–9. These examples reinforce the conclusions drawn in the main text, demonstrating that our method consistently outperforms prior works in standard composition tasks.

Furthermore, we demonstrate the robustness of our model in particularly challenging scenarios. As shown in Figs. 10, Figs. 11 and 12, our method successfully handles compositions involving specified non-trivial shapes, significant object occlusion, and large-angle viewpoint changes. Across all these complex cases, our approach maintains high-fidelity appearance details while achieving precise geometric alignment, showcasing a level of control and quality that existing methods fail to achieve.

**Ablation Study** The results presented in the main text are based on CLIP-derived semantic embeddings. To verify the robustness of the proposed framework across different semantic embeddings, we further validate it using DINO-derived semantic embeddings. As shown in the Tab. 4, our method, based on DINO-derived semantic embeddings, can also effectively capture the geometric properties of objects (see IR and FID metrics), and can successfully retrieve and align appearance features to the corresponding geometric regions based on the captured object geometry (see LPIPS and DISTS metrics). This demonstrates that the proposed framework is robust to different semantic encoders.

Table 4: Quantitative analysis of using different semantic encoders Oquab et al. (2023); Radford et al. (2021).

| Method | IR↑ | FID↓ | LPIPS↓ | DISTS↓ | CLIP↑ | DINO↑ |
|---|---|---|---|---|---|---|
| Ours (CLIP) | 61.14 | 15.04 | 14.94 | 18.53 | 89.38 | 69.92 |
| Ours (DINO) | 61.16 | 14.96 | 14.92 | 18.34 | 89.45 | 69.99 |

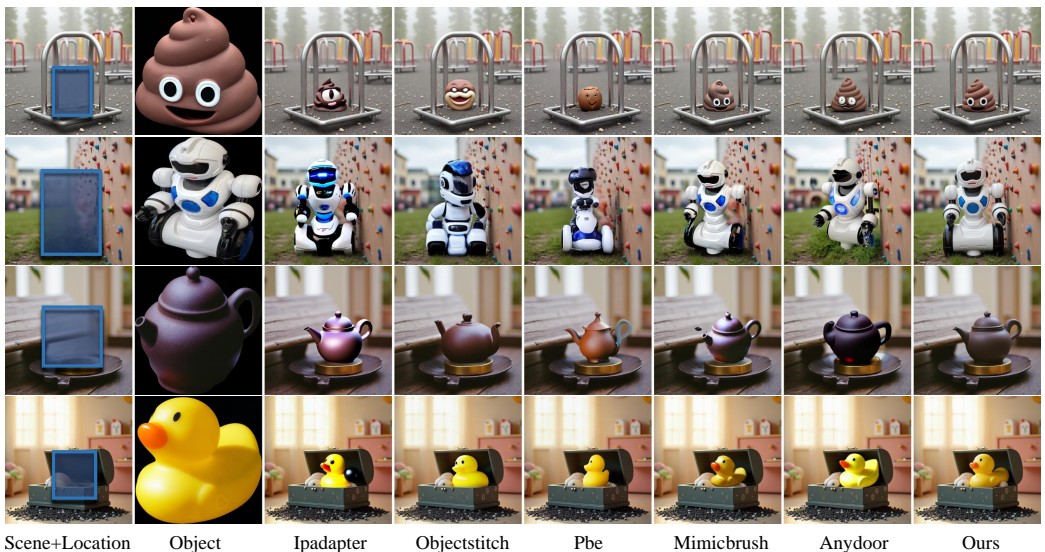

Scene+Location    Object    Ipadapter    Objectstitch    Pbe    Mimicbrush    Anydoor    Ours

Figure 5: Qualitative comparison with recent advanced methods. The images shown are sampled from the test dataset.

## A.3 CODE AND DATASET

To ensure better reproducibility, we have provided the complete code for the proposed framework in a file named `DGAD.zip`, which is included in the supplementary materials. The implementation primarily depends on three core libraries: PyTorch 2.1.1, Transformers 4.51.3, and xformers 0.0.23. Please navigate to the `src` directory and run `pip install -e .` to install the dependencies according to the requirements specified by the `diffusers` library. Once these dependencies are properly installed, the environment required to run the code is fully configured. Regarding the data, the repository contains an `image_data.json` file that specifies the data format used during training. This dataset can be readily obtained following the Anydoor paper Chen et al. (2024). After preparing the data accordingly, the training can be initiated by executing the `train.sh` script. During this process, the necessary pre-trained weights will be automatically downloaded from Huggingface.

## A.4 USE OF LLMs STATEMENT

We utilized Large Language Models (e.g., GPT-4) exclusively for the purpose of polishing the manuscript. Their use was strictly limited to improving grammar, clarity, style, and overall readability. The core scientific ideas, methodologies, experimental design, results, and conclusions were conceived and written entirely by the human authors. No scientific content, code, or experimental designs were generated or influenced by LLMs.

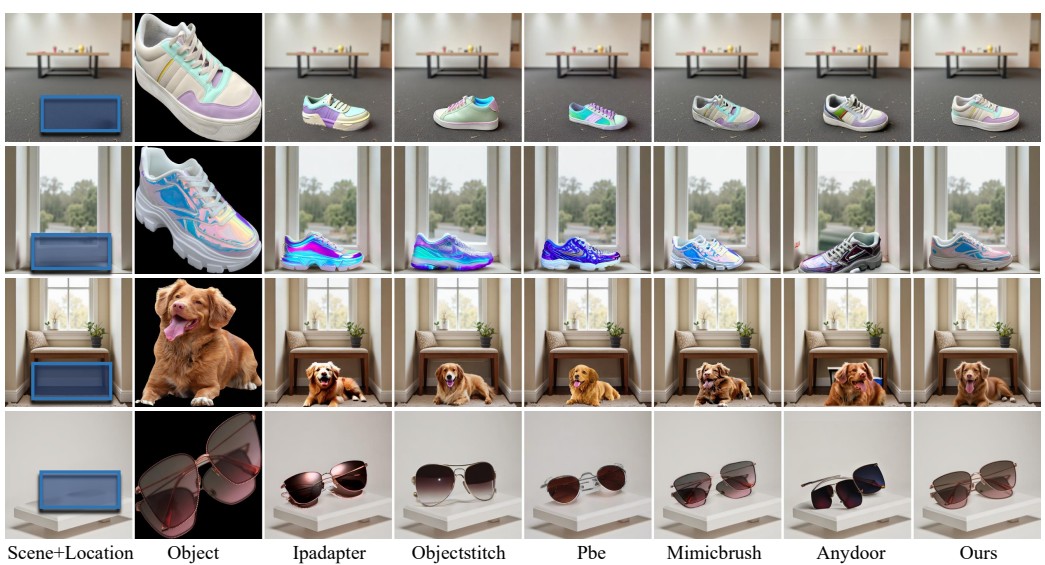

Figure 6: Qualitative comparison with recent advanced methods. The images shown are sampled from the test dataset.

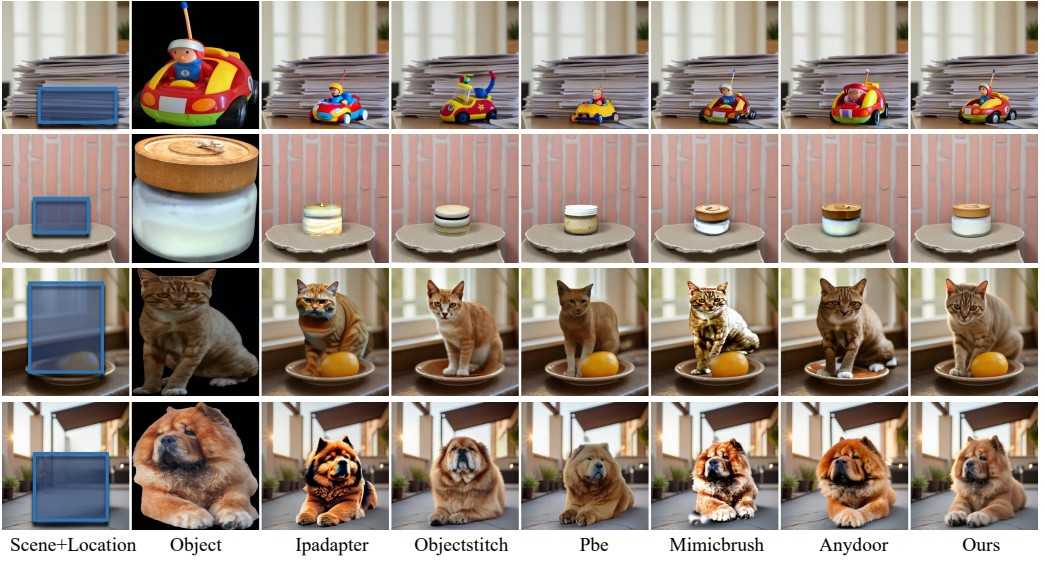

Figure 7: Qualitative comparison with recent advanced methods. The images shown are sampled from the test dataset.

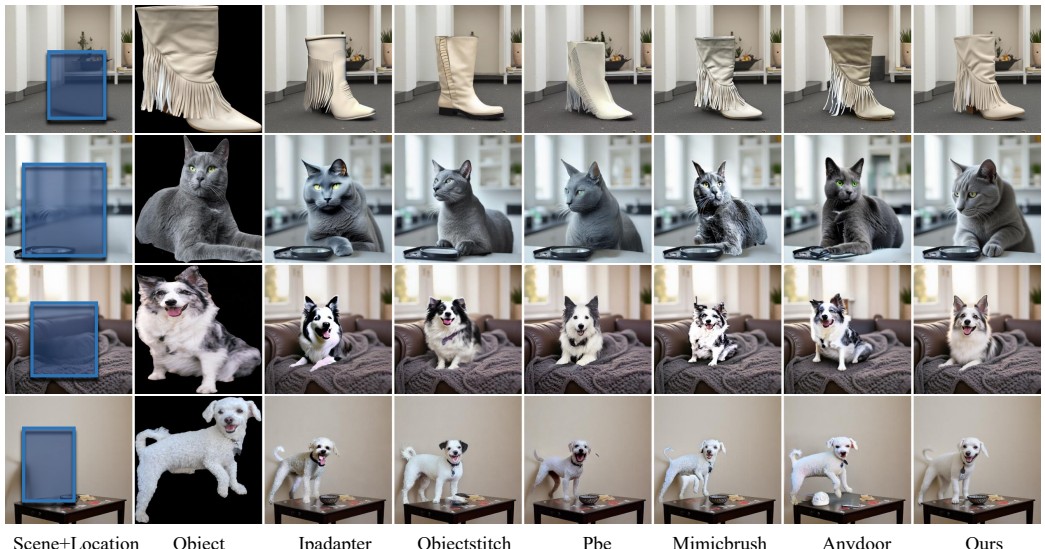

Scene+Location  Object  Ipadapter  Objectstitch  Pbe  Mimicbrush  Anydoor  Ours

Figure 8: Qualitative comparison with recent advanced methods. The images shown are sampled from the test dataset.

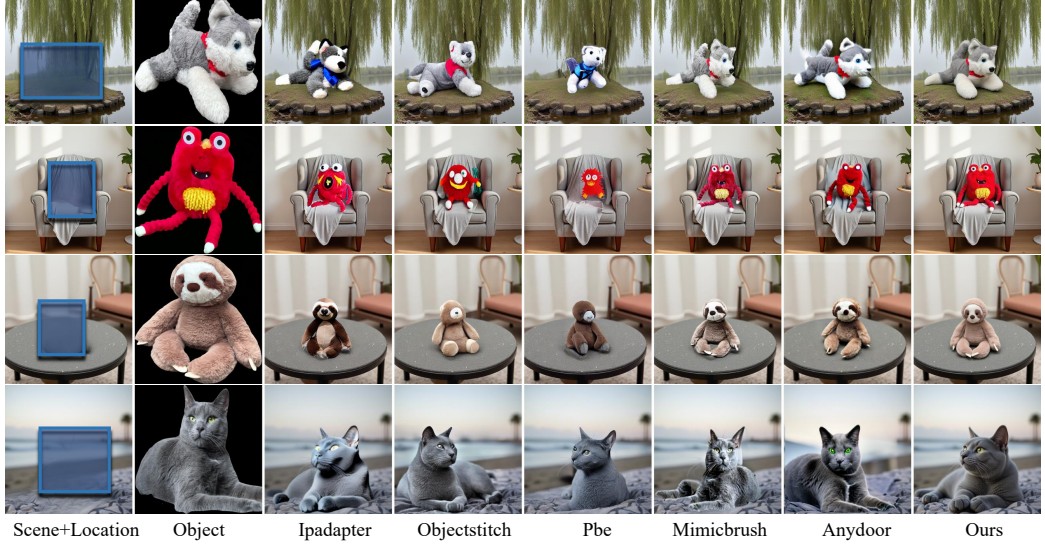

Scene+Location  Object  Ipadapter  Objectstitch  Pbe  Mimicbrush  Anydoor  Ours

Figure 9: Qualitative comparison with recent advanced methods. The images shown are sampled from the test dataset.

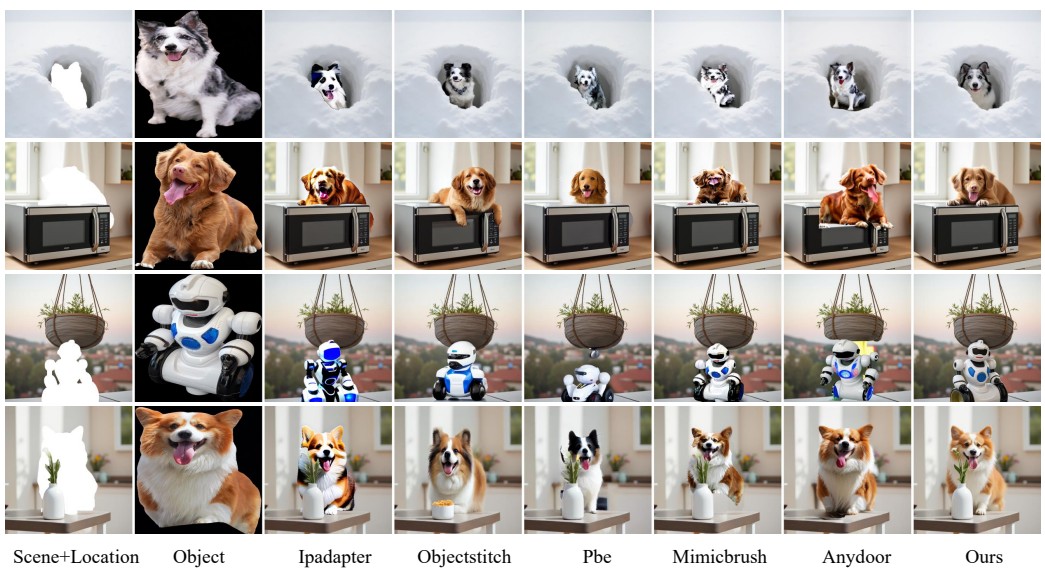

Scene+Location   Object   Ipadapter   Objectstitch   Pbe   Mimicbrush   Anydoor   Ours

Figure 10: Qualitative comparison with recent advanced methods. The images shown are sampled from the test dataset.

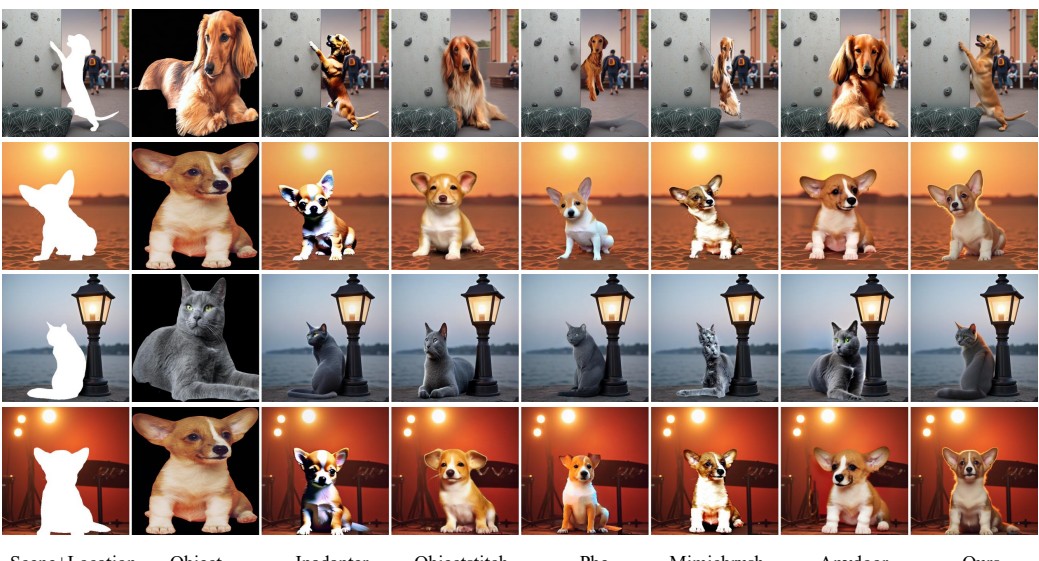

Scene+Location   Object   Ipadapter   Objectstitch   Pbe   Mimicbrush   Anydoor   Ours

Figure 11: Qualitative comparison with recent advanced methods. The images shown are sampled from the test dataset.

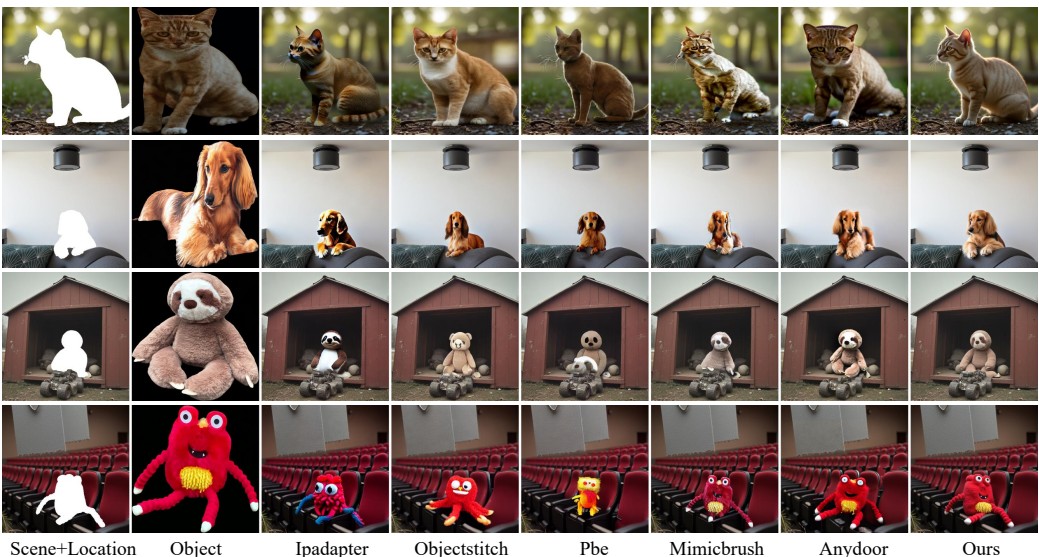

Scene+Location   Object   Ipadapter   Objectstitch   Pbe   Mimicbrush   Anydoor   Ours

Figure 12: Qualitative comparison with recent advanced methods. The images shown are sampled from the test dataset.

