# OpenReview forum: "Geometry-Editable and Appearance-Preserving Object Composition"
_ICLR.cc/2026/Conference — Submitted to ICLR 2026_

### Official Review · Reviewer_rrdB · 2025-10-17

**Soundness:** 2
**Presentation:** 2
**Contribution:** 1
**Rating:** 2
**Confidence:** 3

**Summary:**

This paper proposes Disentangled Geometry-editable and Appearance-preserving Diffusion (DGAD), a diffusion-based approach for object-level customized image generation. The model takes in an object specified by a reference image and places it at a desired location in a background image. The goal is to keep the object fidelity while blending it into the background in a natural way. Prior works in this direction can be mainly categorized into two types: 1) encode objects to semantic tokens (e.g., with DINO or CLIP) and condition the model with cross-attention, 2) train a reference network to extract reference image features and inject to the generator. DGAD is a combination of the two directions -- it uses both cross-attention with sparse semantic object features, and correlation with dense reference network features. Both quantitative and user study results show that DGAD outperforms baselines in object preservation and image quality.

**Strengths:**

- The combination of sparse semantic features and dense reference features is reasonable and turns out to improve performance.
- The ablation on each component of the model is comprehensive.

**Weaknesses:**

1. While combining two previous directions improves results, the technical novelty is still limited. Nothing is surprising to me after reading the paper. The paper writing feels more like a technical report instead of an insightful research paper. For example, can you conduct more analysis on what is learned in the gated dense cross-attention to see if the gating weights indeed highlight the target object?
2. My biggest concern is that the baselines compared in the paper are too old. All these baselines are more than one year ago. Yet we know that personalized image generation or editing is a rapidly evolving field. It feels outdated to experiment on and compare with Stable Diffusion (U-Net) based methods as the community has moved to DiT. Can you compare with some recent approaches such as [1], [2], and [3]? It is hard to assess the effectiveness of the approach without comparing with SOTA methods
3. I doubt if dense cross attention is really that important. Recently works seem to suggest that simple sequence-concat of noisy latents and conditioning tokens + large Transformer is enough to learn pixel correlations [4, 5, 6]. I wonder if the observations in this paper can really transfer.

[1] Chen, Xi, et al. "Unireal: Universal image generation and editing via learning real-world dynamics." CVPR. 2025.

[2] Yu, Qifan, et al. "Anyedit: Mastering unified high-quality image editing for any idea." CVPR. 2025.

[3] Song, Wensong, et al. "Insert anything: Image insertion via in-context editing in dit." arXiv preprint arXiv:2504.15009 (2025).

[4] Tan, Zhenxiong, et al. "Ominicontrol: Minimal and universal control for diffusion transformer." ICCV. 2025.

[5] Labs, Black Forest, et al. "FLUX. 1 Kontext: Flow Matching for In-Context Image Generation and Editing in Latent Space." arXiv preprint arXiv:2506.15742 (2025).

[6] Wu, Chenfei, et al. "Qwen-image technical report." arXiv preprint arXiv:2508.02324 (2025).

**Questions:**

1. Is the dataset the same as AnyDoor's evaluation dataset? The description of the dataset seems to be the same (30 DreamBooth objects + 80 COCO background images), yet the results of AnyDoor seems to be different from their paper.
2. Does DGAD also supports text input? For example, can we specify the pose of the object in the generated image?
3. Please polish the paper writing. For example:
- Please unify the method names across the paper (method names in Tab.1 and the paper text are not consistent in capitalization).
- The spacing between section headers and the text also seems a bit weird.
- In the legend of Fig.2, "Dense Cross-Attention" should be `D` not `S`?

---

### Official Review · Reviewer_5TGd · 2025-10-30

**Soundness:** 3
**Presentation:** 3
**Contribution:** 2
**Rating:** 4
**Confidence:** 4

**Summary:**

This paper proposes DGAD, a diffusion-based framework for “general object composition”: placing an object into a new background with arbitrary pose/placement while preserving its original appearance. The key claim is that previous methods either (a) enable flexible editing but lose fine texture (semantic embedding–driven methods like PBE / ObjectStitch / IP-Adapter) or (b) preserve texture but behave like copy-paste and can’t really reorient or rearticulate the object (appearance-reference methods like MimicBrush). DGAD tries to get both.

The core idea is to explicitly disentangle geometry control and appearance preservation.
- Encoding stage (geometry-editable encoder): The object is encoded to a compact semantic representation (CLIP/DINO), and that is injected via cross-attention into an inpainting-style diffusion UNet that also sees the background latent and an explicit spatial layout prior (mask + context). This stage is supposed to teach the model to implicitly capture editable object geometry (pose, angle, deformation) without requiring dense supervision like full masks at train time for every new edit.
- Decoding stage (appearance-preserving decoder): At decoding time, instead of letting the diffusion model hallucinate appearance, DGAD runs a dense cross-attention module between the geometry-edited latent features and dense appearance features from a reference network (BrushNet). There’s also a spatial gating mask α derived from the geometry features, so that appearance features are only injected where the object should be and don’t leak to background.
- The model is trained end-to-end (keeping SD1.5-inpaint and BrushNet mostly frozen, training new attention modules) on ~386k images + 23k videos processed into triplets (object, background, target composite).
- Experiments: quantitative comparisons across editability metrics (IR, FID), appearance metrics (LPIPS, DISTS), and semantic alignment metrics (CLIP / DINO score); user study; ablations removing either semantic guidance or appearance features; visual comparisons. DGAD reports SOTA on all metrics.

**Strengths:**

1. The method has a plausible inductive bias.
    - Geometry is handled where diffusion models are already good (spatial reasoning inside UNet, with CLIP/DINO guidance and layout conditioning).
    - Appearance is injected later with an explicit dense cross-attention that is spatially gated (α / β masks).
That decomposition feels principled: “coarse where it’s cheap, fine where it matters.” The mask-gated dense attention is more specific than generic “just cross-attend to reference features,” and the ablation suggests it matters (LPIPS/DISTS worsen without dense attention).

2. Empirical results are broad: The authors compare against common baselines (AnyDoor, MimicBrush, IP-Adapter, PBE, ObjectStitch), both numerically and via user study (Composition Quality vs Visual Consistency). There are also ablations that knock out each core piece (semantic guidance, dense attention, layout prior, copied weights) and show consistent drops.

3. Training practicality / reproducibility.
DGAD is trained with 4×4090 GPUs, SD1.5-inpaint backbone, BrushNet reference net, and mostly lightweight cross-attention heads. Releasing code + weights and claiming minimal changes to SD1.5 helps with real-world impact and with ICLR reproducibility.

**Weaknesses:**

### 1. Novelty vs. integration.

DGAD resembles a careful integration of existing ideas rather than a fundamentally new principle. Concretely:
- The geometry-editable encoder looks similar to taking an SD-inpaint UNet and augmenting it with CLIP/DINO features (in spirit, not unlike IP-Adapter / PBE style semantic injection) plus spatial priors reminiscent of AnyDoor.
- The appearance-preserving decoder resembles BrushNet/MimicBrush-like per-pixel reference fusion. DGAD’s main differentiator appears to be the α/β-gated dense cross-attention meant to prevent appearance leakage outside the object region.

One could characterize DGAD as “IP-Adapter for pose + BrushNet for texture + a spatial gating mask,” i.e., incremental engineering. The current draft asserts novelty of the dense gated cross-attention, but does not provide a head-to-head baseline where everything is identical except for that gating. Without such a baseline, it is still plausible that the core gains stem from stitching together two known paradigms, not from a fundamentally new mechanism.

What would help here is a controlled baseline: same geometry encoder + BrushNet-like fusion but without α/β gating, and then quantitative evidence that this baseline either leaks texture into background or fails under strong pose changes. That would isolate DGAD’s core claimed novelty.

### 2. “Implicit geometry” may be overstated.

The paper repeatedly claims DGAD “implicitly captures geometric transformations” by leveraging the diffusion model’s spatial reasoning, rather than requiring explicit 3D supervision. DGAD still takes:
- A spatial layout prior (mask/location);
- A background latent that already encodes where the object should appear;
- Training triplets constructed from compositing procedures (including video-derived pose variation), which explicitly supervise plausible placements.

In other words, DGAD is supervised on how to place the object. The model may not predict full 3D pose or depth, but it is not “free-form reasoning about geometry.” It learns from synthetic pairs how an object can plausibly sit in a scene.

One could say: “This is strong 2D supervision, not emergent geometric understanding.” That weakens the conceptual pitch that DGAD unlocks fundamentally new geometric control.

The paper would benefit from an extrapolation/robustness experiment: show DGAD handling extreme out-of-plane rotations, large scale changes, or deformations unlikely to appear in training triplets, outperforming baselines. That would support the claim of genuine geometric flexibility, not just memorization of common edits.

### 3. Metric interpretation and validity.

The evaluation uses IR, FID, LPIPS, DISTS, and CLIP/DINO similarity and maps them to claims like “editability,” “appearance preservation,” and “semantic consistency.”

Several issues arise:
1. IR is underspecified. The paper refers to IR as reflecting “human preferences” or “editability,” but does not precisely define how IR is computed (which model, what prompts, what aspect it measures). Without that, IR risks sounding like “some black-box heuristic that happened to go up.”
2. FID conflates several factors. A lower FID on the final composite is good, but it mostly says the overall image looks realistic. It does not directly prove correct geometric placement or pose plausibility.
3. LPIPS/DISTS under pose change. LPIPS compares generated object appearance to the reference. But if DGAD rotates the object 60–90° relative to the reference, LPIPS will naturally worsen even if DGAD correctly preserves fine texture and color on visible regions. The paper claims LPIPS/DISTS support “appearance preservation even under geometric transformations,” but does not justify why those view-dependent distortions don’t invalidate the comparison.

Right now, the mapping from metric → claimed property (“editability,” “appearance faithfulness,” etc.) is looser than it should be. The metrics only partially capture the stated property.

The paper would be helped by:
- A precise definition of IR and an explanation of why it correlates with a human notion of correct composition / geometry.
- A demonstrated correlation between IR (and maybe FID) and the user study’s “Composition Quality” score.
- A viewpoint-normalized appearance metric (e.g., identity/texture similarity features designed to be rotation-tolerant) or a human study axis explicitly focused on “how well does the inserted object retain its original markings / texture.”

### 4. Limited discussion of failure modes and robustness.

The qualitative figures in the main paper are mostly successes. There is little analysis, for instance, of (a) cases with thin structures, transparency, or strong specular reflections; (b) scenes requiring consistent lighting/shadow casting; (c) occlusion/ordering issues when the pasted object should go partially behind a foreground element; (d) sensitivity to mask quality. The method appears to assume that at inference time the user (or some upstream step) provides a fairly accurate silhouette / placement region. How robust is DGAD if that mask is approximate, shifted, or slightly too big?

**Questions:**

1.	IR metric: How exactly is IR computed? Which pretrained model or scoring pipeline is used, and why should the community interpret it as “editability” or “composition realism” rather than just “image looks nice”? Please also report its correlation with the human “Composition Quality” score.
2.	Train/test leakage: Are the 30 evaluation objects and 80 backgrounds guaranteed to be unseen during training, at the instance level? Are there near-duplicate frames from the 23k training videos that overlap with those objects, effectively letting DGAD memorize their textures from multiple views?
3.	Novelty of dense cross-attention gating: Can the authors provide a baseline that uses the same geometry encoder and BrushNet-like reference fusion but without α/β spatial gating? In other words, how much of DGAD’s gain is attributable specifically to that gating versus just smartly combining semantic guidance with an appearance-retrieval branch?
4.	Mask robustness / user control: At inference, how precise must the spatial prior (mask / placement) be? Can DGAD take a coarse blob and still infer the right silhouette and pose, or does it require a nearly perfect cutout mask? Please quantify: for example, performance as a function of mask jitter or dilation.
5.	Lighting / shadows / harmonization: Does DGAD ever synthesize shadows consistent with the target scene, or is relighting out of scope? Many of the qualitative examples appear well blended, but it is unclear whether this is due to relighting or simply picking scenes where lighting mismatch is mild.
6.	Runtime and practicality: What is the inference-time cost relative to SD1.5-inpaint alone? Is DGAD realistic for interactive editing, or does it target offline content creation?
7.	Handling of unseen viewpoints: When DGAD rotates an object to reveal surfaces never observed in the source reference, how are those surfaces hallucinated? Are those hallucinated regions still considered “appearance-preserving,” or are they closer to text-guided synthesis? Examples would help.

---

> ### Comment · Reviewer_5TGd · 2025-11-24
> **Look forward to engaging**
>
> Dear authors, I look forward to engaging whenever you get a chance to respond. That said, please prioritize all reviews equally.

---

### Official Review · Reviewer_6PMo · 2025-10-31

**Soundness:** 2
**Presentation:** 3
**Contribution:** 1
**Rating:** 2
**Confidence:** 4

**Summary:**

This paper proposes DGAD, an object insertion framework that claims to be geometry-editable and appearance-preserving. The unique contribution possibly lies in the dense cross-attention mechanism, which proposes a position-wise gating weight to fuse inserted object features with their mask regions. The experiments validate its effectiveness.

**Strengths:**

1. The writing is good. This work proposes the DGAD model for object insertion and presents a plausible way to fuse the object's insertion location and appearance in the dense cross-attention mechanism. The story's claim of "geometry-editable" actually means no distortion of the background, and "appearance-preserving" actually means no distortion of the inserted object.

2. The proposed "dense cross-attention" seems to distribute a dynamic, position-wise gating weight for fusing the query feature (inserted object's CLIP feature) and the key/value features (masks), but it is actually just the mask area, as shown in Equations (5) and (6).

**Weaknesses:**

1.  Many claims in this paper are overstated. The so-called "geometry-editable" feature is only compared to a simple baseline (Figure 1(a), directly inserting a CLIP feature) and does not represent true 3D geometry editing. This is misleading to the reader. A good paper requires a solid contribution, not just a compelling narrative.

2.  Similarly, the "appearance preserving" claim is only compared to methods using a single-branch reference network.

3.  The proposed "dense cross-attention" in Equations (5) and (6) appears to be a re-implementation of latent blending, simply parameterized with dynamic weights, as also validated in Algorithm 1.

4.  The unique contribution appears very limited. Furthermore, the base model is SD 1.5, which is far outdated. The latest methods it is compared against are from CVPR 2024, which are also outdated.

**Questions:**

See weakness

---

### Official Review · Reviewer_s7VW · 2025-11-01

**Soundness:** 2
**Presentation:** 2
**Contribution:** 3
**Rating:** 4
**Confidence:** 3

**Summary:**

This paper proposes the DGAD (Disentangled Geometry-editable and Appearance-preserving Diffusion) model, aiming to address a core challenge in general object composition (GOC): how to preserve the appearance details of an object while editing its geometric properties. The authors decouple the two objectives of geometric editing and appearance preservation by implicitly learning geometric transformations using semantic embeddings during the encoding phase and explicitly aligning appearance features using a dense cross-attention mechanism during the decoding phase.

**Strengths:**

1. The proposed decoupling framework is well designed, handling geometry and appearance separately in the encoding and decoding stages, thus avoiding the overload problem of a single representation. The idea of ​​implicitly capturing geometric attributes using the spatial reasoning capabilities of a pre-trained diffusion model is insightful.

2. Quantitative experiments cover multiple evaluation metrics (editability, appearance preservation, semantic consistency).  The ablation experiments are well-designed, systematically validating the effectiveness of each component.

**Weaknesses:**

1. Technical descriptions are sometimes overly lengthy; core contributions could be expressed more concisely. The system framework diagram in Figure 2 is unclear, making it difficult to quickly understand the overall architecture.

2. The learning process of the position-gated weights $\alpha$ is not sufficiently described

3. There is a lack of failure case analysis, and the limitations and boundary conditions of the method are not discussed. The user research scale is small (only 25 people), resulting in insufficient statistical significance. Comparison of runtime and computational cost is not provided.

4. There is a lack of theoretical explanation for why the decoupled design can simultaneously improve two objectives. The mechanism of implicit geometric learning lacks in-depth exploration. The paper claims to have learned geometric properties "implicitly" through semantic embeddings during the encoding phase, but how can we verify that the model actually captures the correct geometric transformations rather than simply learning statistical patterns in the dataset? Specifically:

- Is the attention map visualization on the left side of Figure 4 sufficient to demonstrate geometric understanding?

- Does the method still work when the geometric transformations of the test object exceed the training distribution (e.g., more extreme viewpoints or deformations)?

- How can we ensure that the position-gated weights α in Dense Attention accurately correspond to the edited object region, rather than the original object region?

**Questions:**

1.  Please provide an analysis of some failure cases, especially under what conditions the method fails? For example, extreme viewpoint changes, severe occlusion, or non-rigid deformation.

2.  How much more computational overhead does the Dense Attention mechanism add compared to standard cross-attention? What is the overall inference time of the method? How does the computational cost compare to the baseline method?

---

### Meta-Review · Area_Chair_xiYv · 2025-12-28

**Summary:**

Reviewer concerns can be grouped as follows:
* Technical novelty. All reviewers noted issues with technical novelty. Several reviewers (6PMo, 5TGd, rrdB) characterized the work as a "careful integration" or "re-implementation" of existing ideas rather than a fundamental breakthrough, while not acknowledging this within the writing.
* Failure case analysis. Several reviewers noted issues in the presentation of the work, specifically a lack of failure case analysis. Highlighting failures would significantly improve the interpretation of the work (s7vw, 5Tgd).
* Baseline comparisons and evaluation framework. Several reviewers noted issues in the evaluation framework, either because the comparisons are too old (comparing with stable diffusion instead of diffusion transformers, and comparing to CVPR 2024 papers as baselines only (rrdB, 6pmo) or because the metrics used do not capture what the paper claims (5Tgd).

These are relatively serious concerns. While there were also strengths that were consistently highlighted (regarding the narrative and presentation in particular), without a rebuttal, the concerns outweigh the strengths, and the paper will therefore be rejected.

**Reviewer Concerns:**

No rebuttal was posted, therefore all reviewer concerns remain outstanding.

**Reviewer Scores:**

No reviewer would change their score since the authors did not post a rebuttal.

---

### Decision · Program_Chairs · 2026-01-26

Reject